# A Photochemically Active Cu_2_O Nanoparticle Endows Scaffolds with Good Antibacterial Performance by Efficiently Generating Reactive Oxygen Species

**DOI:** 10.3390/nano14050452

**Published:** 2024-02-29

**Authors:** Yushan He, Jun Zan, Zihui He, Xinna Bai, Cijun Shuai, Hao Pan

**Affiliations:** 1Hunan 3D Printing Engineering Research Center of Oral Care, Department of Periodontics, Xiangya Stomatological Hospital, Xiangya School of Stomatology, Central South University, Changsha 410008, China; 216811009@csu.edu.cn (Y.H.); 226811005@csu.edu.cn (Z.H.); 2Institute of Additive Manufacturing, Jiangxi University of Science and Technology, Nanchang 330013, China; zanjun5060@163.com; 3Hunan 3D Printing Engineering Research Center of Oral Care, Department of Conservative Dentistry and Endodontics, Xiangya Stomatological Hospital, Xiangya School of Stomatology, Central South University, Changsha 410008, China; baixinna@csu.edu.cn; 4State Key Laboratory of Precision Manufacturing for Extreme Service Performance, College of Mechanical and Electrical Engineering, Central South University, Changsha 410083, China; 5Shenzhen Institute of Information Technology, School of Sino-German Robotics, Shenzhen 518172, China; 6College of Mechanical Engineering, Xinjiang University, Urumqi 830017, China

**Keywords:** porous scaffold, cuprous oxide, reduced graphene oxide, antibacterial, photodynamic therapy

## Abstract

Cuprous oxide (Cu_2_O) has great potential in photodynamic therapy for implant-associated infections due to its good biocompatibility and photoelectric properties. Nevertheless, the rapid recombination of electrons and holes weakens its photodynamic antibacterial effect. In this work, a new nanosystem (Cu_2_O@rGO) with excellent photodynamic performance was designed via the in situ growth of Cu_2_O on reduced graphene oxide (rGO). Specifically, rGO with lower Fermi levels served as an electron trap to capture photoexcited electrons from Cu_2_O, thereby promoting electron-hole separation. More importantly, the surface of rGO could quickly transfer electrons from Cu_2_O owing to its excellent conductivity, thus efficiently suppressing the recombination of electron-hole pairs. Subsequently, the Cu_2_O@rGO nanoparticle was introduced into poly-L-lactic acid (PLLA) powder to prepare PLLA/Cu_2_O@rGO porous scaffolds through selective laser sintering. Photochemical analysis showed that the photocurrent of Cu_2_O@rGO increased by about two times after the incorporation of GO nanosheets, thus enhancing the efficiency of photogenerated charge carriers and promoting electron-hole separation. Moreover, the ROS production of the PLLA/Cu_2_O@rGO scaffold was significantly increased by about two times as compared with that of the PLLA/Cu_2_O scaffold. The antibacterial results showed that PLLA/Cu_2_O@rGO possessed antibacterial rates of 83.7% and 81.3% against *Escherichia coli* and *Staphylococcus aureus*, respectively. In summary, this work provides an effective strategy for combating implant-related infections.

## 1. Introduction

Implant-related infections, as the main complication in implants, have always been one of the most challenging issues in clinical treatment [1,2]. According to reports, implant-related infections account for 60% of bacterial infections [3]. Excessive bacterial infection inevitably increases the healing time, causes incomplete repair of the damaged area, and even leads to the failure of implantable therapy, posing a serious threat to the patient’s life [4]. Currently, antibiotics are often used in clinical practice to resist bacterial infections [5]. However, the excessive use or abuse of antibiotics can not only result in the emergence of bacterial resistance but also have adverse effects on normal tissues [6,7]. Metal-containing materials have been widely used in the antibacterial field; for example, Sousa et al. [8] prepared a peptide-based hydrogel scaffold containing copper (Cu) and found that the scaffold exhibited good antibacterial and rheological properties. Pereira et al. [9] studied the antibacterial effect of composite materials coupled with silver nanoparticles and antibacterial polymers. The results indicated that the composite material has good antibacterial activity, which is higher than that of pure silver nanoparticles. It is worth noting that the excessive release of metal ions can lead to cytotoxicity and ultimately damage normal tissues [10,11]. To overcome the above obstacles, it is particularly important to develop a safe and efficient antibacterial strategy.

Photodynamic therapy (PDT) is considered a promising antibacterial strategy due to its advantages of high selectivity, high efficiency, and minimal side effects [12]. PDT utilizes visible light to activate photosensitizers to produce cytotoxic oxygen species (ROS), which can oxidize bacterial glutathione and proteins, further killing bacteria [13,14]. Moreover, it is important that PDT is unable to lead to the emergence of drug resistance [15]. For example, Zhou et al. [16] designed a new class of HDP-simulated antibacterial compounds and found that the antibacterial activity of the compound not only was three orders of magnitude more potent against bacteria relative to toxicity against RBCs, but also did not induce resistance to 21 sub-inhibitory passages. Among numerous photosensitizers, cuprous oxide (Cu_2_O) is regarded as one of the photocatalysts with the greatest potential because of its narrow band gap and low toxicity [17]. Xia et al. [18] prepared a Cu_2_O/MXene composite structure that exhibited desirable synergistic antibacterial function in response to an infection microenvironment under light irradiation.

However, the narrow band gap of Cu_2_O leads to the rapid recombination of photogenerated electrons and holes, thereby restricting its photocatalytic efficiency [19]. For example, Wang et al. [20] prepared an antibacterial Cu_2_O/g-C_3_N_4_ composite and found that Cu_2_O/g-C_3_N_4_ could generate ROS to effectively kill bacteria under light irradiation. Nevertheless, the narrow band gap of Cu_2_O leads to the rapid recombination of the photogenerated electrons and holes, thereby restricting its photocatalytic efficiency.

Reduced graphene oxide (rGO), as a graphene material, has attracted a lot of research in the field of photocatalysis, ascribed to its unique two-dimensional nanostructure, excellent electron mobility, and huge specific surface area [21,22,23]. Kayode et al. combined rGO with ZnS and utilized this combination’s electron-capture and -transfer capabilities to enhance its ability to separate photoexcited carriers from ZnS composite materials, thus achieving superior optoelectronic applications [24]. Taking inspiration from the above, a composite of graphene oxide and Cu_2_O is expected to be a promising way to enhance photoelectric performance [25]. Firstly, rGO serves as an electron trap to capture photoexcited electrons from Cu_2_O, thus inhibiting electron-and-hole recombination in Cu_2_O. Secondly, the excellent conductivity of rGO could quickly transfer electrons from Cu_2_O, further suppressing the recombination of electron-hole pairs. In this study, Cu_2_O@rGO nanoparticles were first prepared via the in situ growth of Cu_2_O on the surface of rGO. Subsequently, Cu_2_O@rGO nanoparticles were introduced into poly-L-lactic acid (PLLA) porous scaffolds via additive manufacturing technology. The morphology, crystal structure, and chemical state of Cu_2_O@rGO nanoparticles were thoroughly analyzed. The optical properties of Cu_2_O@rGO nanoparticles were comprehensively evaluated through photoluminescence spectroscopy, EIS, and transient photocurrent response. Additionally, the photogenerated ROS ability and Cu ion release concentration of the scaffold were detected. Furthermore, the antibacterial properties against *Escherichia coli* and *Staphylococcus aureus* were evaluated in depth. Eventually, the biocompatibility of the scaffold was also evaluated.

## 2. Materials and Methods

### 2.1. Material Source

PLLA powder (with a melting point of ~170 °C) was purchased from Shenzhen Boli Biomaterials Co., Ltd. (Shenzhen, China). Graphene Oxide (GO) nanosheet was provided by Chengdu Organic Chemicals Co., Ltd., of the Chinese Academy of Sciences (Chengdu, China). Cu_2_O nanoparticles, cupric acetate monohydrate (Cu(Ac)_2_·H_2_O, 99%), sodium hydroxide (NaOH) and ascorbic acid were procured from Aladdin Biological Technology Co., Ltd. (Shanghai, China).

### 2.2. Preparation of Cu_2_O and Cu_2_O@rGO

The Cu_2_O nanoparticles were loaded onto the surface of GO using a facile hydrothermal method (Figure 1). Specifically, 0.4 g of Cu(Ac)_2_·H_2_O was dissolved in 50 mL of distilled water and continuously stirred at room temperature. Afterwards, GO nanosheets were dispersed in 50mL of aqueous solution to form an aqueous dispersion, which was then blended with the aforementioned solution and stirred continuously for 30 min. Then, 50 mL of NaOH (0.2 mol/L) and 30 mL of ascorbic acid (0.1 mol/L) were gradually added to the above suspension and stirred for 30 min at 50 °C to form a Cu_2_O@rGO composite. Ultimately, Cu_2_O@rGO nanoparticles could be obtained through filtration, washing with deionized water and alcohol, centrifugation, and drying at 45 °C.

### 2.3. Scaffold Fabrication

The Cu_2_O@rGO nanoparticles and PLLA powder were loaded into a separate beaker containing 150 mL of anhydrous ethanol, and then, stirred for 30 min. Subsequently, the two dispersions were merged and ultrasonically stirred for 2 h. Immediately, the PLLA/Cu_2_O@rGO powder could be obtained through centrifugation, drying, and grinding. Eventually, PLLA/Cu_2_O@rGO porous scaffolds were prepared via selective laser sintering (SLS), as shown in Figure 1. The corresponding process parameters were as follows: scan speed of 240 mm/s, laser power of 1.5 W, and layer thickness of 0.1 mm. The surface morphology of PLLA/Cu_2_O and PLLA/Cu_2_O@rGO porous scaffolds is shown in Figure 1b. Clearly, the color of PLLA/Cu_2_O scaffolds was pink, whereas the color of PLLA/Cu_2_O@rGO scaffolds was black, which was ascribed to the incorporation of GO nanosheets. By SEM observation, it was seen that the PLLA/Cu_2_O@rGO scaffold exhibited a relatively good density and its pore size was ~500 um. Additionally, the hydrophilicity of the bracket was detected by a water contact angle instrument, as shown in Figure 1b. As is well known, the PLLA scaffold was a hydrophobic polymer, whereas the PLLA/Cu_2_O@rGO scaffold exhibited relatively hydrophilic properties, as evidenced by its water contact angle of <90°. The results indicate that the scaffold was more easily adhered to by cells or bacteria, allowing them to function better [26,27].

### 2.4. Characterization of Nanoparticles

The morphology of Cu_2_O and Cu_2_O@rGO nanoparticles was analyzed through transmission electron microscopy (TEM, JEOL, 2100F, Akishima, Japan). The chemical structure of the samples was analyzed by a Raman system (Finder Vista, Zolix Instrument Co., Ltd., Beijing, China). Meanwhile, the crystal structure composition was detected by X-ray diffraction (XRD, DY2472, Empyrean, Eindhoven, The Netherlands) with a scanning angle range of 5–90°, scanning speed of 5°/min, and voltage of 40 kV. Eventually, the chemical state was measured via X-ray photoelectron spectroscopy (XPS, ESCALAB Xi+, Thermo Fischer, Waltham, MA, USA). The transient fluorescence spectrometer Photoluminescence (PL) spectra of Cu_2_O and Cu_2_O@rGO were studied using a spectroscopy system of steady-state photoluminescence with 325 nm excitation (Edinburgh FLS1000, UK). An electrochemical workstation (CHI660E, CH Instruments Inc., Shanghai, China) was used to detect the electrochemical properties of the scaffolds. Additionally, an instantaneous photocurrent was performed on the on/off time period from electrode to lamp. Cyclic voltammetry (CV) tests were carried out at a scanning rate of 10 mV/s within a voltage range of −1.25 to 1.25 V (vs. SCE) in a 0.1 M NaOH solution condition.

### 2.5. Antibacterial Experiment

Staphylococcus aureus (*S. aureus*, ATCC49775) and Escherichia coli (*E. coil*, ATCC25922) were selected as test microorganisms for detecting the antibacterial activity of PLLA, PLLA/Cu_2_O, and PLLA/Cu_2_O@rGO scaffolds. Specifically, prior to the antibacterial experiment, the dishes and media were sterilized using high-pressure sterilization pots, and all scaffolds were also sterilized using ultraviolet radiation for 1 h. Subsequently, the bacteria and the scaffolds were co-cultivated for 24 h at 37 °C. Next, the composites were illuminated by visible light for 20 min, with a light distance of 40 cm. Afterwards, we took out the cultivated bacteria-scaffold complexes and washed them with PBS around three times. Thereafter, the bacteria on the surface scaffold was shaken for 10 min, and then, the bacterial turbid solution was diluted to 1 × 10^9^ CFU/mL. Moreover, 150 μL of the diluted bacterial solution was added to LB agar plates and cultivated at 37 °C for 12 h. Finally, the number of colonies was counted using a digital camera and Image J software (version 1.8.0) [28,29]. Additionally, the antibacterial rate was further calculated using the following equation:Antibacterial rate (%) = [1 − (CFU sample/CFU control)] × 100%.
where CFU control represents the bacteria number without any treatment and CFU experimental indicates the bacteria number cultured on the PLLA/Cu_2_O@rGO scaffold. The morphology of the bacteria cultured on the scaffolds was studied using SEM devices. In short, all scaffolds were cultivated with bacterial solution (1 × 10^8^ CFU/mL) at 37 °C for 24 h. After the scheduled cultivating time, all experimental groups were irradiated with or without light for 30 min. Subsequently, the bacteria-scaffold complexes were taken out and repeatedly washed three times using PBS. Then, the bacteria-scaffold complexes were fixed with glutaraldehyde (2.5%) for 1 h. Afterwards, the dehydration process was conducted. Ultimately, the morphology of the bacteria-scaffold complexes was observed through a scanning electron microscopy device (SEM, Zeiss Gemini 300, Oberkochen, Germany). Live-dead staining images of bacteria on the scaffolds were detected via a live-dead staining agent (calcein-AM and propidiumiodide).

### 2.6. ROS Detection

The extracellular ROS level produced via the PLLA/Cu_2_O and PLLA/Cu_2_O@rGO scaffolds under light-irradiation conditions was detected via a methylene blue (MB) degradation experiment, and the ROS content was recorded through UV–visible absorption and fluorescence spectroscopy. Specifically, PLLA/Cu_2_O and PLLA/Cu_2_O@rGO were first added to MB solution (40 μg/mL), and then, cultivated in the dark for 20 min. Then, 100 μL solutions were taken out from the above plates and placed into a multifunctional microporous reader, and we recorded the absorbance from 550 nm to 750 nm. Additionally, the generation of singlet oxygen (^1^O_2_) by PLLA/Cu_2_O and PLLA/Cu_2_O@rGO scaffolds was evaluated using 1,3-Diphenylisobenzofuran (DPBF). Briefly, all scaffolds were placed into a 6-well plate containing anhydrous ethanol (2.9 mL) and DPBF (100 μL) for detecting ^1^O_2_, and then, the corresponding absorbance of the solutions was recorded from 350 nm to 500 nm via a multifunctional microplate reader (Xlement SPR100, Liangzhun Industrial Co., Ltd., Shanghai, China).

The intracellular ROS levels inside the bacteria generated by various scaffolds were detected using 2′,7′-Dichlorodihydrofluorescein diacetate (DCFH-DA, Solarbio) as a capture agent, due to DCFH-DA being easily oxidized into green DCF. Specifically, after the bacterial culture was completed, the bacterial suspension (1 × 10^7^ CFU/mL) was transferred to an EP tube, followed by the placement of the scaffolds. After 30 min of irradiation or darkness treatment, all scaffolds were removed, and then, the bacterial suspension was washed two times with Phosphate-Buffered Saline (PBS) solution, as well as centrifuged to obtain bacteria. Next, DCFH-DA (200 μL, 1 mol/L) capture agent was mixed with the aforementioned bacteria, and then, incubated at 37 °C for 30 min under dark conditions. After that, the above solution was centrifuged and washed three times with PBS, using PBS (500 uL) to dilute it. Ultimately, green fluorescent images were obtained through fluorescence microscopy, and the DCF fluorescence intensity was analyzed by ImageJ software (version 1.8.0).

### 2.7. Protein Leakage Assessment

Bacterial integrity could be determined by assessing protein leakage within the bacteria, which is usually detected by a Coomassie Brilliant Blue Kit (G250). In detail, the bacteria were co-cultured with various scaffolds for 12 h in the absence or presence of light excitation. Next, all scaffolds were taken out and centrifuged for 3 min to collect the supernatant. Immediately, 400 μL G250 capture agent was blended with 100 μL supernatant and cultivated for 10 min at 37 °C. Ultimately, the protein leakage level of the bacteria solution was evaluated by a multifunctional microplate reader to record the absorbance at 562 nm.

### 2.8. Cell Behavior

Mouse bone marrow mesenchymal stem cells (mBMSCs) were chosen as a model to analyze the biocompatibility of various scaffolds. Specifically, mBMSCs were cultured with a DMEM medium containing 10% fetal bovine serum in 5% carbon dioxide and 37 °C environments, and we replaced the culture medium at 1-day intervals. When the cell density reached about 85%, a trypsin solution was used to digest cells for expansion and culture until the number of cells met the demand. Before co-cultivation, all scaffolds were irradiated with ultraviolet light for 1 h in anhydrous alcohol, and subsequently cleaned three times via PBS. Then, the cells (density 1 × 10^4^ cell/mL) were co-cultured with the aforementioned scaffolds, and then, placed in a 24-well plate in 5% carbon dioxide and 37 °C environments. Subsequently, all groups were treated with light or darkness after 3 and 5 days of cultivation. After a specific time, the cells co-cultured with the scaffolds were washed three times with PBS and marked for cultivation for 30 min via a live-dead staining agent (calcein-AM and propidiumiodide (PI)) at room temperature. Finally, the stained cells were observed using a fluorescence microscope (BX51, Olympus, Japan).

Additionally, the cell proliferation effect promoted by scaffolds was evaluated via the Cell Counting Kit-8 (CCK-8, Solarbio, Beijing, China) method. In detail, after the set cultivation time, the fresh medium with CCK-8 solution (10%) was dropped onto all groups, and then, co-cultivation was carried out at 37 °C for 1 h in a dark environment. Eventually, the absorbance of the above suspensions at 450 nm was detected by a multifunctional microplate reader. Subsequently, to further elucidate the release mechanism of Cu ion, the Cu ion concentration released from the PLLA/Cu_2_O and PLLA/Cu_2_O@rGO scaffold at various periods was quantitatively evaluated through inductively coupled plasma emission spectroscopy (ICP-OES). In detail, all scaffolds (∅6 × 2 mm) were immersed in a scaffold/PBS ratio of 1.0 g/50 mL for different durations (0, 1, 2, 3, 4, 5, 6, 7, 8 h). Then, the scaffolds were removed and the soaking liquid (0.5 mL) was absorbed 10 times. Finally, the Cu^2+^ concentration in the absorbed liquid (1 mL) was tested by lCP-OES analysis. Additionally, the released Cu ions in each time period were accumulated for evaluating the total Cu ion release concentration.

### 2.9. Statistical Analysis

All data were tested three times and the corresponding results are presented as mean ± standard deviation. The statistical differences were estimated via Student’s *t*-test, in which *p* < 0.05 (*), *p* < 0.01 (**), and *p* < 0.001 (***) were considered to indicate statistical significance.

## 3. Results and Discussion

### 3.1. Microstructure of the Synthesized Nanoparticles

The morphology of Cu_2_O@rGO nanoparticles was observed using TEM, and the obtained results are shown in Figure 2a. Obviously, some spherical nanoparticles grew on the GO nanosheet, and their average size is about 73 nm (Figure 2(a_1_)). To verify the components of the nanoparticles, a more in-depth analysis was conducted on the red box area, as shown in Figure 2(a_2_). According to a fast Fourier transform (FFT) calculation, the lattice spacing of the nanoparticles was 0.23 nm, which belonged to the (111) crystal plane of Cu_2_O (Figure 2(a_3_)) [30]. Meanwhile, the electron diffraction images show that the nanoparticles possessed three crystal planes, which were assigned to the (111), (110), and (220) crystal planes of Cu_2_O (Figure 2(a_4_)) [31]. To sum up, there is reason to believe that these nanoparticles on nanosheets belonged to expectant Cu_2_O.

XRD was used to analyze the phase structure of the nanoparticles, as shown in Figure 2b. Obviously, GO had a sharp diffraction peak at 11.6°, which corresponds to its typical (001) crystal plane [32]. As for Cu_2_O@rGO nanoparticles, some characteristic peaks at 29.7°, 36.5°, 42.4°, 61.4°, and 73.6° were observed, which were attributed to the (110), (111), (200), (220), and (311) crystal planes of Cu_2_O, respectively [33]. The diffraction peak of rGO could not be found, which might be due to the fact that the insertion of Cu_2_O led to a decrease in the interlayer spacing of graphite and prevented GO re-accumulation [34]. Raman spectra were carried out to verify the effective reduction of GO into rGO, as shown in Figure 2c. The analysis showed that the intensity ratios (*I*_D_/*I*_G_) in Cu_2_O@rGO (1.03) are far higher than those in GO (0.91), suggesting the formation of more sp^3^ defects [35,36,37,38].

Furthermore, XPS spectroscopy was used to analyze the electronic chemical state of Cu_2_O@rGO, as shown in Figure 2d–f. The whole XPS spectrum of Cu_2_O@rGO possessed Cu, O, and C electronic orbitals, which were basically consistent with those of pure Cu_2_O. Furthermore, the Cu_2_O spectrum showed that the strong characteristic peaks symbolizing Cu^+^ 2p_3/2_ and Cu^+^ 2p_1/2_ were located at 932.38 and 952.38 eV, whereas the characteristic peaks symbolizing Cu ion 2p_3/2_ and Cu ion 2p_1/2_ were located at 934.68 and 955.08 eV, respectively [39]. As a comparison, both peak positions symbolizing Cu^+^ in Cu_2_O@rGO redshifted, which indicates that there is an interaction between rGO and Cu_2_O [40]. Based on previous studies, a reasonable explanation is that Cu^+^ utilized oxygen-containing functional groups of rGO as targets for the in situ growth of Cu_2_O [41,42].

### 3.2. Photodynamic Performance of Cu_2_O@rGO

To understand the enhancement in photodynamic activity and the recombination and separation behavior of electron-hole pairs (e^−^−h^+^) in Cu_2_O@rGO, the steady-state photoluminescence spectrum (PL), transient photocurrent response spectrum, and electrochemical impedance spectroscopy (EIS-Nyquist) were comprehensively characterized. The PL spectrum characterized the separation efficiency of e^−^ and h^+^, and higher intensity represented lower separation efficiency [43]. As shown in Figure 3a, it is clear that the PL emission peak of Cu_2_O@rGO was lower than that of Cu_2_O, indicating its better efficiency of charge transfer and its better ability to prevent the electron-hole recombination [44,45].

The charge transfer efficiency of the samples was detected by transient photocurrent response spectra, as shown in Figure 3b. Obviously, the maximum photocurrent of Cu_2_O under illumination did not exceed 2 μA, whereas the photocurrent of Cu_2_O@rGO could reach ~3.32 μA. The higher photocurrent density represented a better photoelectric effect [46]. Additionally, the electric resistance was also evaluated via EIS-Nyquist, as shown in Figure 3c. Generally, a larger of diameter of the EIS-Nyquist arc represents greater the resistance during electron transfer [47]. The arc diameter of Cu_2_O@rGO was significantly lower than that of pure Cu_2_O, indicating that electrons were easier to transport on the Cu2O@rGO surface. The photoelectric stability of Cu_2_O@rGO was evaluated using cyclic voltammetry (CV), as shown in Figure 3d. During ten potential cycles from −1.25 V to 1.25 V, the current density for Cu_2_O@rGO exhibited no obvious attenuation, indicating that it possessed excellent cyclic stability under light conditions.

Based on the results above, it is clear that Cu_2_O@rGO possessed good photodynamic activity, with the corresponding mechanism shown in Figure 3e. As is well known, the Fermi potential level of rGO was −4.42 eV, whereas the conduction band (CB) of Cu_2_O was −1.14 eV [48,49]. In this case, the photoinduced electron (e^–^) preferentially transferred from Cu_2_O to rGO. More importantly, the surface of rGO could quickly transfer electrons from Cu_2_O due to its excellent conductivity, thus efficiently suppressing the recombination of electron-hole pairs. The photoexcited electrons and holes induced the excess accumulation of intracellular ROS, such as superoxide anions (·O^2–^) or hydroxyl radicals (·OH).

### 3.3. In Vitro ROS Detection

Methylene blue (MB) and 1,3-Diphenylisobenzofuran (DPBF) were selected for ROS detection, aiming to explore the types and capabilities of ROS generated by the scaffolds [50,51]. Specifically, the MB capture agent was used to evaluate ·OH produced by the scaffolds since MB could be oxidized by ·OH and reduce its absorption peak at 660 nm (Figure 4a). For the PLLA scaffold, the MB absorption peak almost exhibited no changes, implying that it could not produce ·OH under light illumination (Figure 4b). As a comparison, the MB absorption peak for the PLLA/Cu_2_O group only slightly decreased with the increase in light treatment time, which was ascribed to the faint photodynamic effect of Cu_2_O (Figure 4c). Interestingly, as for PLLA/Cu_2_O@rGO, the absorption peak sharply decreased after prolonged light time, and the generation of ·OH by the scaffold was time-dependent, demonstrating its outstanding ·OH generation capacity under light conditions (Figure 4d). Moreover, the DPBF solution was used to detect the production of ^1^O_2_ because DPBF could be oxidized by ^1^O_2_ and reduce its absorption intensity at 410 nm (Figure 4a). As shown in Figure 4e–g, the PLLA scaffold was unable to produce ^1^O_2_ under light conditions, whereas the PLLA/Cu_2_O scaffold displayed a slight ability to produce ^1^O_2_. As for PLLA/Cu_2_O@rGO, the difference was that the absorption peak of DPBF in the PLLA/Cu_2_O@rGO group reduced with increasing time, indicating that it has an excellent ability to photogenerate ^1^O_2_.

### 3.4. Antibacterial Activity of the Scaffold

The antibacterial activity of the scaffolds was evaluated using the agar plate method, and *Staphylococcus aureus* (*S. aureus*) and *Escherichia coli* (*E. coli*) were selected as experimental bacteria [52,53]. As shown in Figure 5a, the number of *S. aureus* cultured on all scaffolds was almost the same as that of blank group under dark conditions, indicating their negligible antibacterial performance. On the contrary, under visible light irradiation, PLLA/Cu_2_O and PLLA/Cu_2_O@rGO exhibited different degrees of antibacterial efficacy, with evidence of a decrease in bacterial number. Similarly, there were no obvious differences in *E. coli* number between the scaffolds and the blank group under dark conditions (Figure 5c). Under visible light irradiation, PLLA/Cu_2_O and PLLA/Cu_2_O@rGO could effectively kill bacterial, which was ascribed to their good PDT effect.

Quantitative analysis was carried out for the number of bacterial colonies, as shown in Figure 5b,d. Obviously, the antibacterial rate of all scaffolds was less than 10% against *S. aureus* and *E. coli* under dark conditions. As a comparison, the antibacterial rates of the PLLA/Cu_2_O group against *S. aureus* and *E. coli* were 49.7% and 51.8% under light irradiation, respectively. Particularly, the antibacterial rates of the PLLA/Cu_2_O@rGO group against *S. aureus* and *E. coli* exceeded 80%, which was attributed to the enhanced photodynamic effect after the incorporation of rGO.

To further observe the state of bacteria, the morphology and membrane integrity of *S. aureus* and *E. coli* on the scaffolds were observed using SEM. The *S. aureus* cultured on PLLA scaffolds displayed typical spherical morphology whether in the absence or presence light-irradiation conditions (Figure 6a). For PLLA/Cu_2_O scaffolds, the bacterial morphology underwent slight deformation under light irradiation, which indicated PDT therapy could effectively destroyed membrane integrity of bacteria. As for PLLA/Cu_2_O@rGO scaffold, the bacterial morphology severely deformed owing to the enhanced PDT effect.

Subsequently, the microscopic morphology of *E. coli* was also analyzed, as shown in Figure 6b. Under dark conditions, the bacteria adhered on the scaffold maintained its integrated strip shape, indicating the scaffolds were difficult to inactivate bacterial. Under light conditions, the bacteria structure on the PLLA/Cu_2_O scaffold were showed some morphological changes. For PLLA/Cu_2_O@rGO scaffolds, the rod-shaped structure of bacterial was wholly damaged. These results proved that PLLA/Cu_2_O@rGO scaffolds possessed excellent antibacterial ability under light irradiation.

### 3.5. Antibacterial Mechanism of the Scaffold

In this study, the DCFH-DA was used to evaluate the ROS content produced by the scaffold [54]. Detail, DCFH-DA could be hydrolyzed into DCFH, which could be further oxidized to green fluorescence of DCF, and its fluorescence intensity could reflect the ROS level in the cell. As shown in Figure 7a,b, a small amount of green fluorescence appeared for various scaffolds in the absence of light irradiation, whereas a relatively large quantity of green fluorescence was observed for PLLA/Cu_2_O and PLLA/Cu_2_O@rGO scaffolds under light-irradiation conditions. The results exhibited that PLLA/Cu_2_O and PLLA/Cu_2_O@rGO could release ROS for antibacterial therapy when stimulated by visible light. Significantly, the amount of green fluorescence for the PLLA/Cu_2_O@rGO scaffold was more than that for the PLLA/Cu_2_O scaffold, indicating its more excellent ROS generation capability. The live-dead staining images of bacteria on the scaffolds are showed in Figure 7e. Obviously, whether it was *S. aureus* and *E. coli*, the majority of bacteria on the PLLA scaffold were dyed green, whereas only a small fraction of the bacteria were dyed red. The results showed that the PLLA scaffold exhibited no antibacterial properties. On the contrary, the majority of bacteria on the PLLA scaffold were dyed red, indicating that the scaffold possessed good antibacterial properties. As is well known, green fluorescence arises from an ROS reaction in cells, indicating dead/dying or affected cells. Hence, according to the ROS assay results and the live-dead staining images, there is reason to believe that the green fluorescence in Figure 7a,b overwhelmingly arises from a ROS reaction in the cells, indicating dead cells. 

Corresponding fluorescence intensity was also detected, as shown in Figure 7c,d. As expected, there was no significant change in the fluorescence levels for all scaffolds under dark conditions. On the contrary, the fluorescence intensity for the PLLA/Cu_2_O and PLLA/Cu_2_O@rGO scaffolds was enhanced after light treatment, and the PLLA/Cu_2_O@rGO scaffold exhibited the highest fluorescence intensity. These results mean that the incorporation of Cu_2_O@rGO could significantly enhance the photogenerated ROS ability of the scaffold, thereby achieving effective antibacterial activity.

As is well known, ROS caused gradual disintegration of the cell membrane, and then, damage to nucleic acid and the leakage of intracellular protein, thus inactivating bacteria. After the bacterial structure was damaged, some important components inside the bacteria would leak out. To this end, G250 was used to evaluate the protein leakage levels within *S. aureus* and *E. coli*. Specifically, G250 could form noncovalent complexes with amino groups, thereby interrupting protein cleavage and causing color changes. Meanwhile, the OD values of the G250 solution at 562 nm could reflect the protein leakage level of bacteria, and a higher OD value indicated more protein leakage. Obviously, the OD value against *S. aureus* for the PLLA group remained unchanged with increasing light exposure time (Figure 7f), indicating that protein leakage could be ignored. In contrast, the OD value for the PLLA/Cu_2_O scaffold increased from 0.11 to 0.16 after lighting for 30 min, whereas the OD value for the PLLA/Cu_2_O@rGO scaffold further increased to 0.41. Similarly, the OD value against *E. coli* for the PLLA/Cu_2_O@rGO scaffold was higher than for the other scaffolds, and the OD value reached a maximum of 0.39 after lighting for 30 min (Figure 7g). The above results clearly confirm that the PLLA/Cu_2_O@rGO scaffold led to serious protein leakage in bacteria through the generation of numerous ROS, which was consistent with the bacterial morphology (Figure 6).

### 3.6. Cellular Activity

For implants, both excellent antibacterial performance and good biocompatibility are needed for evaluating their potential applications. In this work mBMSCs were selected as sample cells for biological experiments, mainly attributed to the fact that they are a kind of pluripotent stem cell with self-renewal and multi-directional differentiation potential. Additionally, mBMSCs possess the characteristics of being a convenient source and exhibiting easy expansion and no immune rejection [55,56]. Therefore, the cytotoxicity of various scaffolds was thoroughly assessed via live-dead staining and CCK-8 assays [57]. Generally, calcein-AM (green fluorescence) and propidium iodide (PI, red fluorescence) reagents were used to stain live cells and dead cells, respectively. As shown in Figure 8a, the majority of cells cultured on the scaffolds were stained green after culturing for 3 days, and the number of green cells further increased after culturing for 5 days. These results imply that all scaffolds provided a comfortable environment for mBMSC growth, thus exhibiting good biocompatibility. The CCK8 analysis method was used to detect the number of cells, and the number of live cells was proportional to the Optical Density (OD) value at 450 nm. Obviously, the OD values for all scaffolds increased with the extension of incubation time (Figure 8b). Moreover, the OD values for the PLLA/Cu_2_O@rGO scaffold after cultivation for 3 and 5 days were 1.3 and 2.3, respectively, which were close to the OD values for the PLLA scaffold. These results reveal that the PLLA/Cu_2_O@rGO scaffold exhibited no side effects in response to cell proliferation and differentiation.

Furthermore, the Cu ion release kinetic curves of the PLLA/Cu_2_O and PLLA/Cu_2_O@rGO scaffolds were investigated, as shown in Figure 8c. Obviously, the Cu ion concentrations released by the PLLA/Cu_2_O and PLLA/Cu_2_O@rGO groups were 72.68 μg/L and 66.80 μg/L in the first 4 h. As the degradation time of the scaffolds increased, the cumulative release of Cu ions slowly rose. After 8 h of degradation, the Cu ion concentration released by the PLLA/Cu_2_O and PLLA/Cu_2_O@rGO groups reached 89.26 μg/L and 96.98 μg/L, respectively. According to relevant reports, cell cytotoxicity is negligible and the side effects on the human body are minimal below a concentration of 1000 μg/L of Cu ions. Hence, it is clear that the Cu ion concentration released by PLLA/Cu_2_O@rGO scaffold was within the safety threshold, which was also an important basis for ensuring biocompatibility.

## 4. Conclusions

In this study, a visible light-responsive Cu_2_O@rGO nano-system was designed, aiming to endow PLLA scaffolds with good antibacterial effects. Specifically, Cu_2_O@rGO nanocomposites were formed through the in situ growth of Cu_2_O on the surface of rGO. Subsequently, Cu_2_O@rGO nanocomposites were mixed with poly (L-lactic acid) (PLLA) powder to fabricate PLLA/Cu_2_O@rGO porous scaffolds via SLS. Photochemical analysis confirmed the PLLA/Cu_2_O@rGO scaffold exhibited excellent ROS generation ability, which contributed to positive photogenerated-electron migration and the separation of electrons from holes. The antibacterial results indicated that the PLLA/Cu_2_O@rGO scaffold effectively killed bacteria under light irradiation, with sterilization rates of 81.3% and 83.7% for *S. aureus* and *E. coli*, respectively. The antibacterial mechanism of the scaffold was achieved by disrupting the structural integrity of bacteria and inducing protein leakage. Furthermore, the PLLA/Cu_2_O@rGO scaffold had good biocompatibility with mBMSCs. Therefore, this study possesses broad application prospects in the field of photodynamic antibacterial therapy.

## Figures and Tables

**Figure 1 nanomaterials-14-00452-f001:**
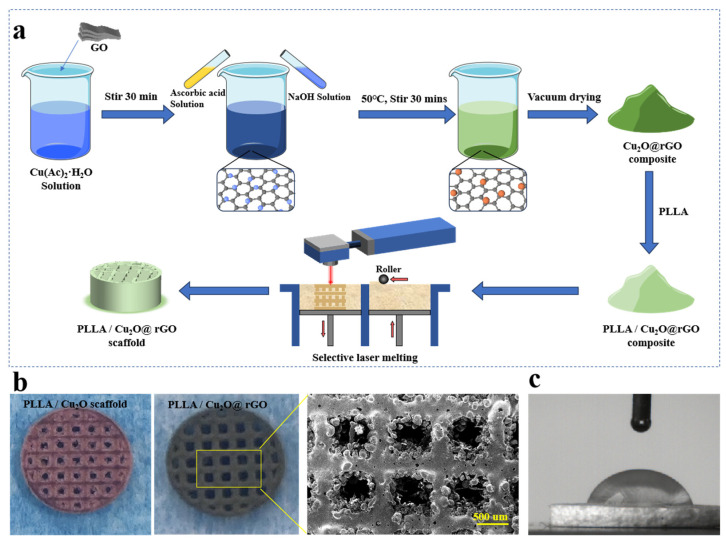
(**a**) The preparation process of Cu_2_O@rGO nanoparticles and PLLA/Cu_2_O@rGO scaffold; (**b**) the surface morphology of the porous scaffolds; (**c**) the hydrophilic angle of the PLLA/Cu_2_O@rGO scaffold.

**Figure 2 nanomaterials-14-00452-f002:**
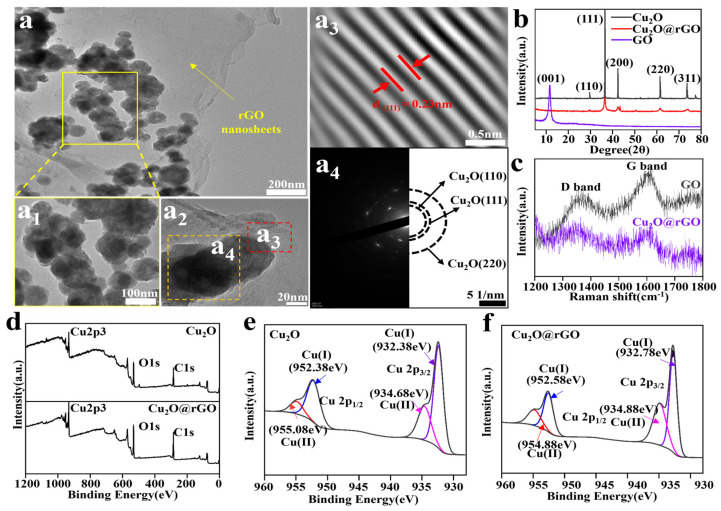
(**a**) TEM images of Cu_2_O@rGO; (**a_1_**,**a_2_**) high rate of the images; (**a_3_**) Fourier transform images of Cu_2_O in Cu_2_O@rGO; (**a_4_**) selected-area electron diffraction (SAED) spectra of Cu_2_O in Cu_2_O@rGO; (**b**) XRD results of Cu_2_O, GO, and Cu_2_O@rGO nanoparticles; (**c**) Raman spectra of GO and Cu_2_O@rGO; (**d**) XPS full spectra of Cu_2_O and Cu_2_O@rGO; (**e**,**f**) fitted curve for XPS results of Cu 2p of Cu_2_O and Cu_2_O@rGO.

**Figure 3 nanomaterials-14-00452-f003:**
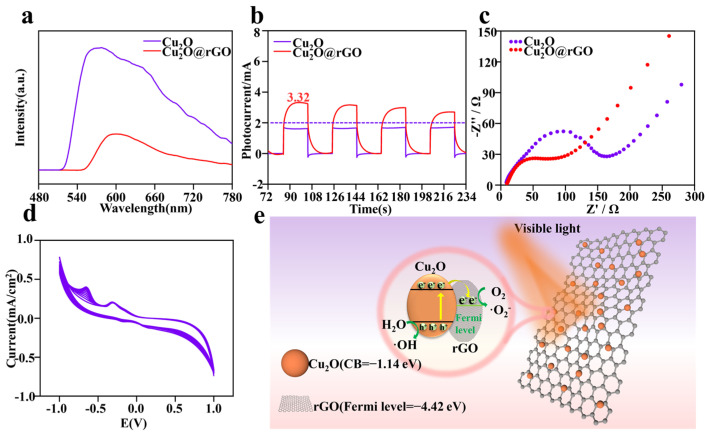
(**a**) Photoluminescence emission spectra of Cu_2_O and Cu_2_O@rGO in the range of 480 to 780 nm; (**b**) four-cycle transient response photocurrent curves of Cu_2_O and Cu_2_O@rGO under light irradiation; (**c**) EIS–Nyquist curve of Cu_2_O and Cu_2_O@rGO in the presence of light excitation; (**d**) ten-cyclic voltammetry cycles of Cu_2_O@rGO in NaOH solution; (**e**) schematic diagram of the mechanism of ROS production of Cu_2_O@rGO nanocomposites.

**Figure 4 nanomaterials-14-00452-f004:**
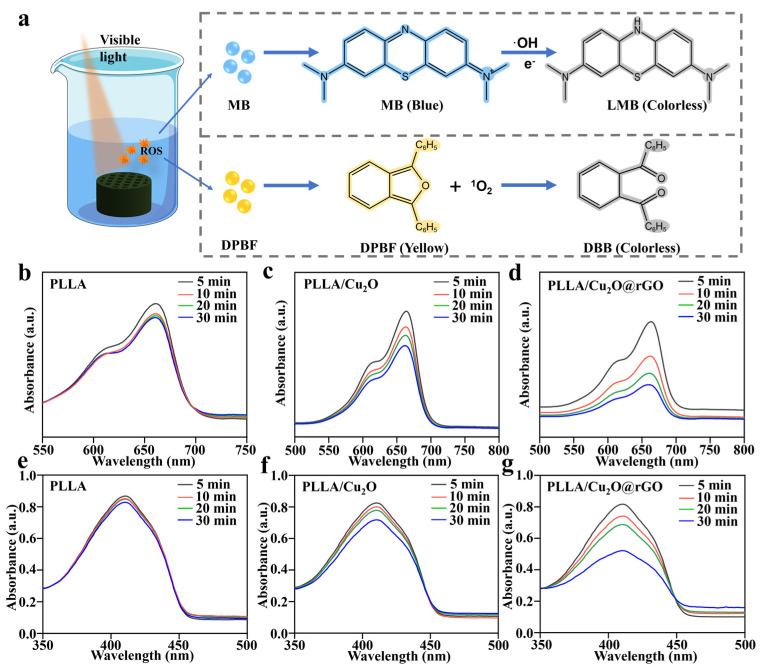
(**a**) Reaction mechanism diagram of extracellular ROS generation; the absorption catalytic effect of MB solution treated with (**b**) PLLA, (**c**) PLLA/Cu_2_O, and (**d**) PLLA/Cu_2_O@rGO scaffolds under light irradiation for 5, 10, 20, and 30 min; the loss level of the DPBF solution immersed with (**e**) PLLA, (**f**) PLLA/Cu_2_O and (**g**) PLLA/Cu_2_O@rGO scaffolds with different light irradiation times (5, 10, 20, and 30 min).

**Figure 5 nanomaterials-14-00452-f005:**
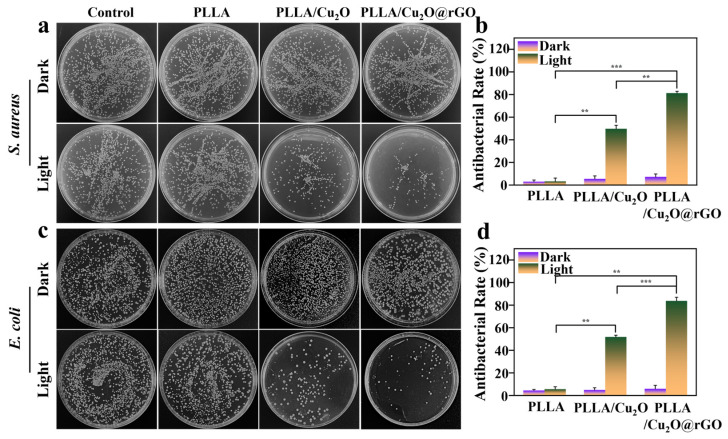
Agar diffusion plate images of (**a**) *Staphylococcus aureus* and (**c**) *Escherichia coli* colony incubated with different scaffolds; the antibacterial rates of the scaffolds against (**b**) *S. aureus* and (**d**) *E. coli. p* < 0.01 (**) and *p* < 0.001 (***).

**Figure 6 nanomaterials-14-00452-f006:**
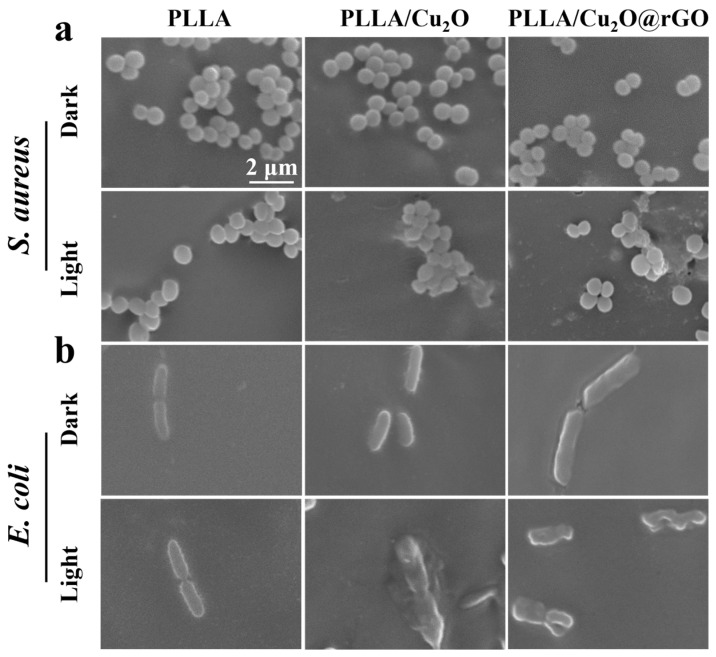
SEM morphology images of (**a**) *S. aureus* and (**b**) *E. coli* treated by various scaffolds with or without light irradiation.

**Figure 7 nanomaterials-14-00452-f007:**
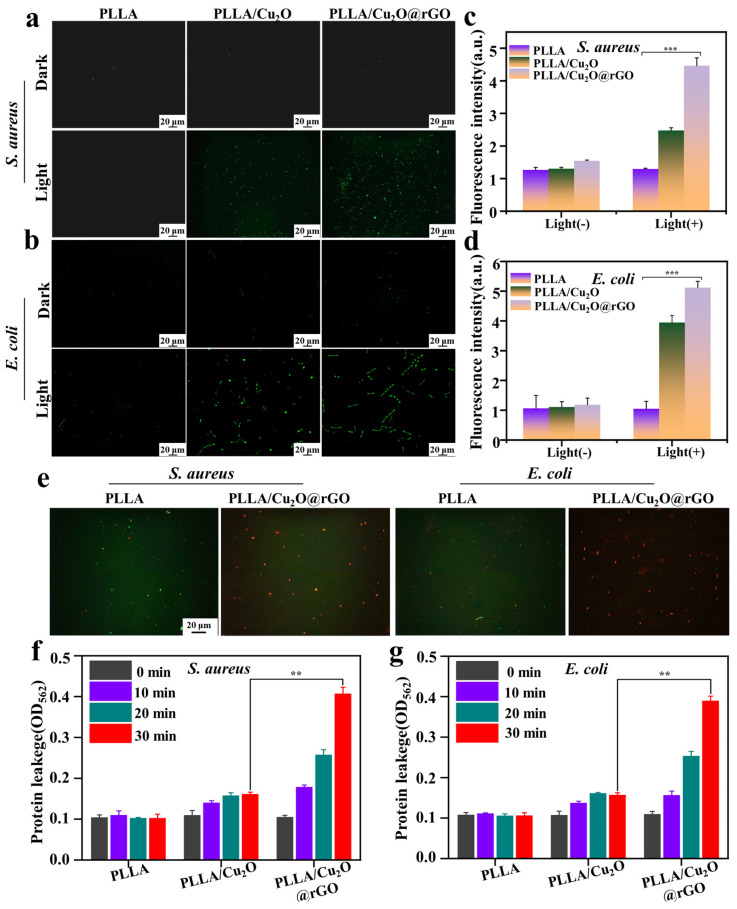
Green fluorescence images of (**a**) *S. aureus* and (**b**) *E. coli*; fluorescence intensity of DCF of (**c**) *S. aureus* and (**d**) *E. coli* on different scaffolds in the absence or presence of light irradiation; (**e**) live-dead staining images of bacteria on the scaffold; the OD_562_ values of G250 solution of (**f**) *S. aureus* and (**g**) *E. coli* cultured on the scaffolds. *p* < 0.01 (**), and *p* < 0.001 (***).

**Figure 8 nanomaterials-14-00452-f008:**
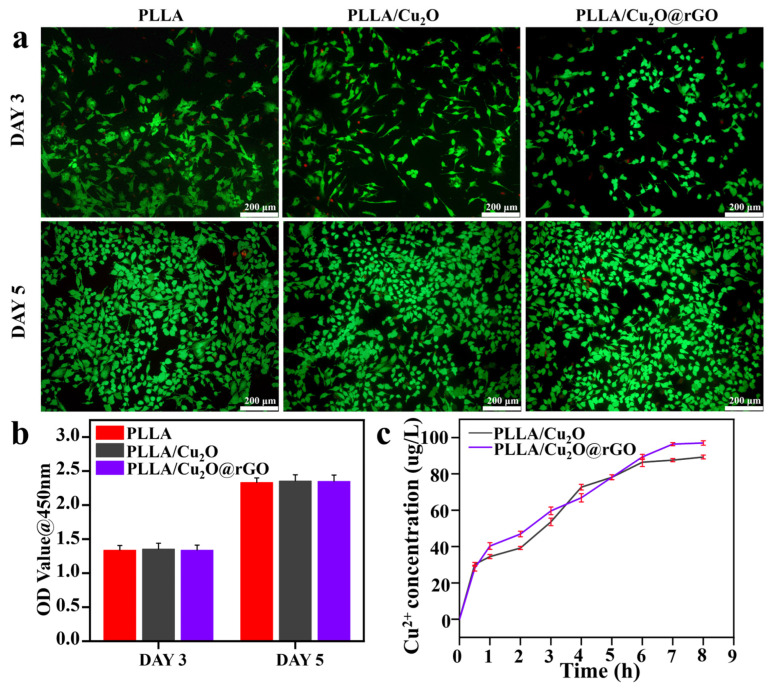
(**a**) The cell live-dead fluorescence staining results of PLLA/Cu_2_O and PLLA/Cu_2_O@rGO scaffolds (green fluorescence represents live cells and red fluorescence represents dead cells); (**b**) CCK-8 assay results after co-culture with different scaffolds for 3 and 5 days with light irradiation. (**c**) Cu ion release kinetic curves of PLLA/Cu_2_O and PLLA/Cu_2_O@rGO scaffolds.

## Data Availability

Data are available upon request.

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
