# Peer review of "A Photochemically Active Cu2O Nanoparticle Endows Scaffolds with Good Antibacterial Performance by Efficiently Generating Reactive Oxygen Species"

_nanomaterials, 2024, doi:10.3390/nano14050452_

Round 1
Reviewer 1 Report
Comments and Suggestions for Authors
The manuscript "A Cu2O@rGO nanocomposite particles endows photodynamic 2 antibacterial performance of the scaffold" describes a novel material that releases Cu ions into solution as an antimicrobial platform. The results are novel and impactful.
Despite the novelty, there are a number of concerns with the manuscript that must be addressed prior to publication
1. The authors MUST do a more thorough introduction to metal-containing and metal-releasing materials for antimicrobial applications. Several starting points for this include (doi: 10.1186/s40580-022-00309-7 ; doi: 10.1002/jbm.a.37677 ; doi: 10.3390/ijms25021256 ; DOI: 10.3390/molecules22091487)
2. The authors should include more background on the efficacy of light-activated antimicrobials and photodynamic approaches. Some starting points include (https://doi.org/10.1021/acsami.8b19098; doi: 10.5978/islsm.27_18-RA-01 ; doi: 10.1016/j.ijbiomac.2023.127685; doi: 10.1021/acsami.3c16335)
3. Throughout the manuscript, the authors refer to Cu2+. How was speciation determined? If not, then the references should be changed to Cu ions or some similar means of description,
4. In figure 7, the green fluorescence arises from ROS-reaction in cells indicating dead/dying or affected cells. Was counter-staining performed to show where the living cells were? I would assume there are living cells in the dark portions of FIgure 7A, but the authors did not present visualization of the cells.
5. The authors only briefly mention the mBMSC cell type used in Figure 8. A short description of what they are is warranted beyond the methods section.
6. How was copper ion concentration determined in Figure 8? I the preparation of samples for ICP-OES is important, and warrants more detail in the methods.
7.Overall the manuscript could use proofreading with a number of awkward english-language phrases/terms. There are also some likely errors in translation (e.g. line 182 - microporous? likely should be microplate). As well as general use of "flowery" language (e.g. "delightedly" line 323
Comments on the Quality of English Language
Needs some improvement. Clearly some inappropriate phrasing and translations.
Reviewer 2 Report
Comments and Suggestions for Authors
In this article, a new nanosystem (Cu2O@rGO) with excellent photodynamic performance was designed via the in-situ grown of Cu2O on reduced graphene oxide (rGO). Subsequently, the Cu2O@rGO nanoparticle was introduced into poly-L-lactic acid (PLLA) powder to prepare PLLA/Cu2O@rGO porous scaffolds through selective laser sintering. Photochemical analysis showed that Cu2O@rGO could enhance the efficiency of photogenerated charge carriers and promote electron hole separation. Here are some comments for further improvement of the article.
Better add the full formulation and specific applicability in the title.
Add some characterization results in the abstract.
I suggest citing some articles on copper oxide and antibacterial application of inorganic nanoarchitectures in the introduction, Adv. Healthcare Mater. 2024, 2303582,
In schematic, it is suggested to place the composite rather than power for better understanding. All section headings should be in lower case except first letter.
poor characterization of final formulation, lacking the surface morphology and physicochemical attributes.
Check all abbreviations and avoid typological errors.
Comments on the Quality of English Language
Line 205 various period? change to various periods.
In such a way, several instances, typos and grammatical errors remained.
For examples to For example.
Line 79 change was to were.
Figure legend: prepared process to preparation process
Round 2
Reviewer 1 Report
Comments and Suggestions for Authors
The authors have improved the manuscript. I Feel the author's description of the live/dead cells regarding my original comment on Figure 7 should be included in the text of the article. You should note in the text (A) the extent of bacterial reduction and (B) that live cells were not visualized. It is an important point as to not unintentionally misread the readers or cause misinterpretation.
Comments on the Quality of English Language
n/a
